# Automating multimodal microscopy with NanoJ-Fluidics

Pedro Almada[1,2], Pedro M. Pereira [1,2,3], Siân Culley [1,2,3], Ghislaine Caillol[4], Fanny Boroni-Rueda[4], Christina L. Dix[1], Guillaume Charras[5,6], Buzz Baum[1,6], Romain F. Laine[1,3], Christophe Leterrier [4] & Ricardo Henriques [1,2,3]

Combining and multiplexing microscopy approaches is crucial to understand cellular events, but requires elaborate workflows. Here, we present a robust, open-source approach for treating, labelling and imaging live or fixed cells in automated sequences. NanoJ-Fluidics is based on low-cost Lego hardware controlled by ImageJ-based software, making high-content, multimodal imaging easy to implement on any microscope with high reproducibility. We demonstrate its capacity on event-driven, super-resolved live-to-fixed and multiplexed STORM/DNA-PAINT experiments.

[1] MRC-Laboratory for Molecular Cell Biology, University College London, London WC1E 6BT, UK. [2] Department of Cell and Developmental Biology, University College London, London WC1E 6BT, UK. [3] The Francis Crick Institute, London NW1 1AT, UK. [4] Aix Marseille Université, CNRS, INP UMR7051, NeuroCyto, Marseille 13015, France. [5] London Centre for Nanotechnology, London WC1H 0AH, UK. [6] Institute for the Physics of Living Systems, University College London, London WC1E 6BT, UK. These authors contributed equally: Pedro Almada and Pedro M. Pereira. Correspondence and requests for materials should be addressed to R.F.L. (email: r.laine@ucl.ac.uk) or to C.L. (email: christophe.leterrier@univ-amu.fr) or to R.H. (email: r.henriques@ucl.ac.uk)

Fluorescence microscopy is ubiquitously used to observe cellular processes thanks to its ease of use, exquisite sensitivity and molecular specificity. It is generally performed using dedicated sample preparation procedures, tailored to achieve optimal imaging conditions for each chosen technique. Furthermore, each technique entails a compromise between temporal/spatial resolution and innocuity to living cells[1]. Unique insights can also be gained by combining information from multiple approaches, but at the cost of complex correlative workflows[2]. Recent developments toward molecular imaging of a large number of targets have introduced the use of multiple rounds of labelling and imaging[3,4]. Additionally, event-driven experiments, where sample treatment is triggered by imaging cues, is proving powerful to study dynamics phenomenon such as mitosis[5]. However, the adoption of such elaborate protocols is commonly hampered by low reproducibility and throughput, limiting their appeal for quantitative work.

Automated fluid handling using microfluidic chips presents an attractive alternative, but adds constraints on culturing conditions and sample preparation[6]. A simple and tractable method would automate fluid exchange in commonly used open imaging chambers, while being easily adaptable to existing microscope. For this, we devised a user-friendly, open-source system called NanoJ-Fluidics (Fig. 1a, b). This automated computer-controlled syringe pump array can reliably exchange fluids at the sample to perform fixation, labelling and imaging (Fig. 1c and Supplementary Fig. 1), making complex multimodal imaging protocols highly accessible to researchers.

## Results

**The NanoJ-Fluidics framework**. NanoJ-Fluidics is a complete system that uses off-the-shelf components and open-source control software. It allows labelling and treatment protocols traditionally done at the bench to be performed automatically and directly on the microscope stage (Supplementary Fig. 1). The hardware consists of compact Lego syringe pumps (Fig. 1a) that can be configured as a multiplexed array of up to 128 units (Fig. 1b), plus a peristaltic pump and an Arduino® controller interface (Fig. 1b). Affordable, low tolerance Lego parts allow pump-based protocols to be robust and repeatable. The system is easy to set up and use (Supplementary Note 1), highly modular and compatible with most microscopes and experimental workflows (Supplementary Fig. 1) and does not require any

microfabrication process as it uses common labware (Supplementary Fig. 2). We designed specific workflows depending on the desired protocol and the volumes of reagents accessible to the researcher (Supplementary Note 2 and Supplementary Fig. 4a). The software is provided as an ImageJ/μManager plugin[7] or as a stand-alone package for independent fluidics control (Supplementary Software 1) for precise control of each steps in the protocol (Supplementary Fig. 3).

In order to challenge the capabilities of our approach and guide in the choice of workflows, we have characterised the accuracy and precision of the volumes provided by NanoJ-Fluidics in a variety of conditions, e.g. across different Lego syringe pumps, syringes and injected volumes (Supplementary Note 3 and Supplementary Fig. 4). In all the performed characterisations using calibrated pumps, both the precision (standard deviation of the error) and accuracy (mean of the error) were below 5% of the nominal injected volume. These high precisions and accuracies combined with appropriate workflows make NanoJ-Fluidics a robust tool to achieve automation of most imaging protocols.

**Event-driven fixation imaging**. NanoJ-Fluidics has the advantage of allowing sample treatments, such as fixation, at precise times during the experiment. Thanks to the integration of NanoJ-Fluidics with the image acquisition, determining the time of treatment can be triggered by imaging cues. To demonstrate this capacity, we carried out an experiment observing the state of focal adhesions, as mammalian cells progress into division. Fixation was triggered by the observation of the rounding of the cells as they approach mitosis[8]. Also, in order to fully exploit the fluidics automation of NanoJ-Fluidics, we combined it with tiling imaging and image stitching in order to obtain fields-of-view of several millimetres while preserving high resolution.

We first blocked asynchronous cells in G2 via treatment with a CDK1 inhibitor[9] (RPE1 cells expressing zyxin-GFP). Next, the cell cycle was released by exchanging the inhibitor by growth media using NanoJ-Fluidics (Fig. 2a) and imaged by live-cell timelapse microscopy (Fig. 2b). When enough cells were observed to undergo mitotic rounding (determined by visual inspection), the fixative was injected (Fig. 2a, b). A considerable portion of cells (~10%) were found to be fixed in a similar rounding state due to the simultaneous release of the G2 block (insets of Fig. 2c I, Supplementary Movie 1). The sample was then immunolabelled for β1-integrin, plus co-stained for F-actin and DNA, then

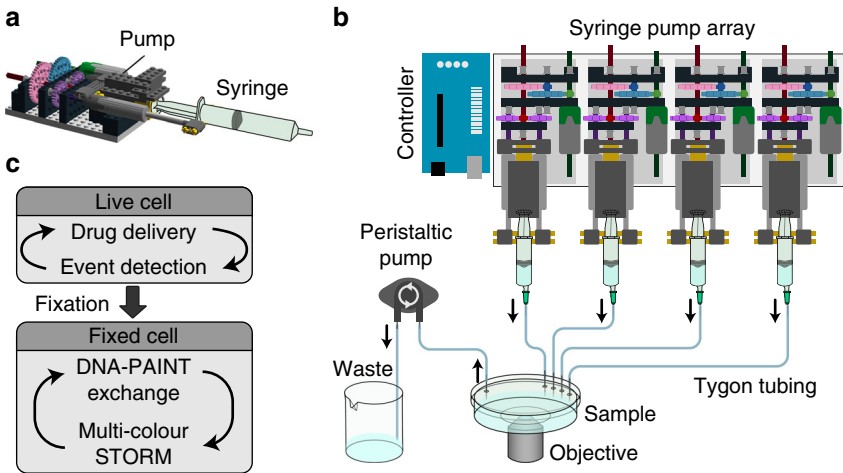

**Fig. 1** Schematics of the NanoJ-Fluidics system. **a** 3D side view of a single syringe pump. **b** 2D top view of a syringe pump array (representing 4 pumps out of 128 maximum) and a fluid extraction peristaltic pump, both controlled by an Arduino UNO. **c** Example of possible workflows

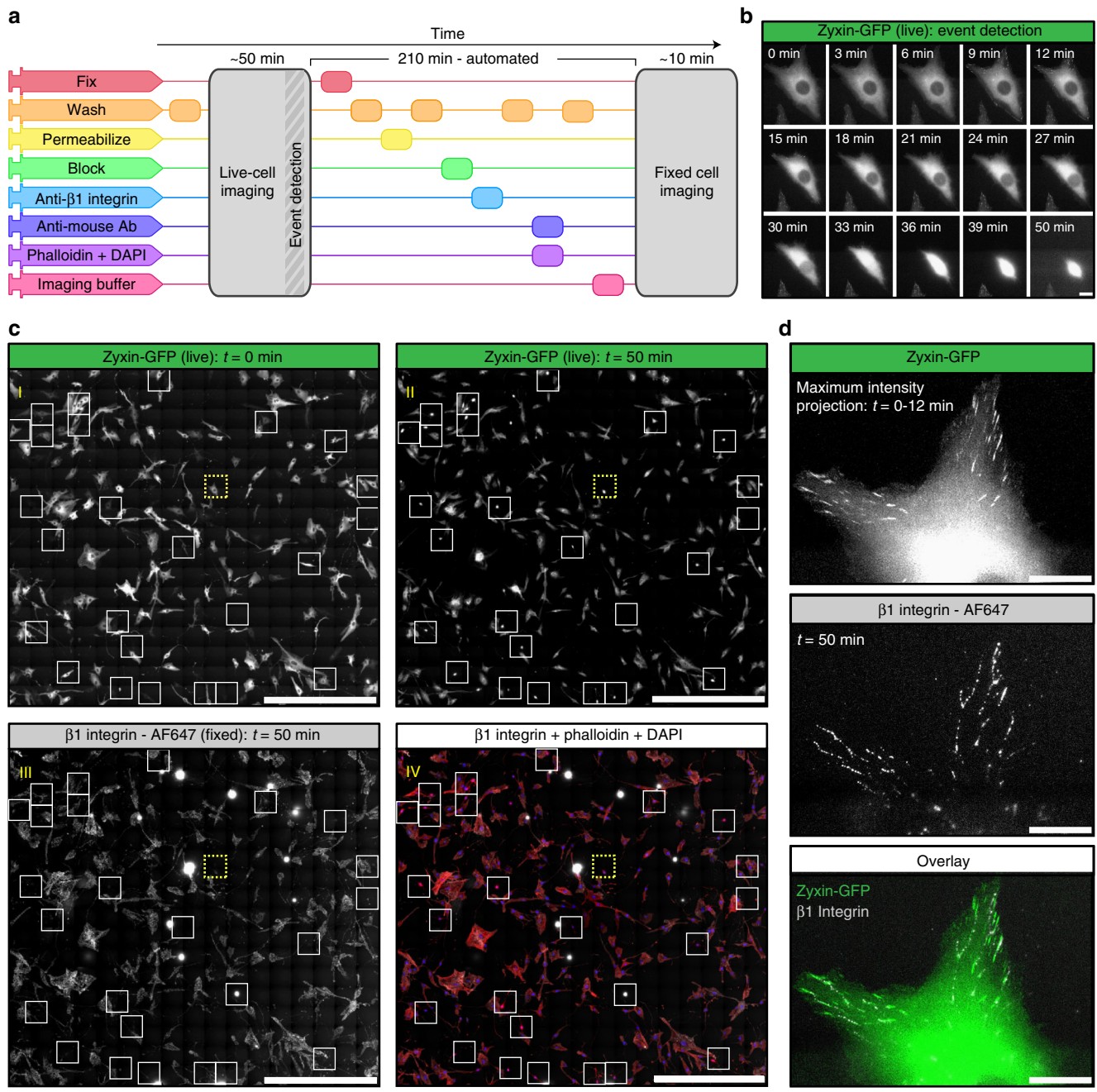

**Fig. 2** Event-driven fixation of cells upon mitotic rounding. **a** NanoJ-Fluidics workflow of the event-driven protocol performed. **b** Stills of RPE1 zyxin-GFP live-cell timelapse during mitotic rounding. Scale bar, 20 μm. **c** Stitched mosaic (17 × 17 individual regions) of: I—First frame of the live-cell timelapse; II—Last frame of the live-cell timelapse; III—RPE1 zyxin-GFP cells immunolabelled for active β1-integrin; IV—Overlay of RPE1 zyxin-GFP cells immunolabelled for active β1-integrin and stained for F-actin (with phalloidin-TRITC) and DNA (with DAPI). White boxes represent cells where mitotic rounding was observed (Supplementary Movie 4), yellow dashed inset is the cell in (**b**) and (**d**). Scale bar is 1 mm. **d** I—Maximum intensity projection of the first 12 min in (**b**); II—Active β1-integrin staining; III—Overlay of both panels. Scale bar, 20 μm

imaged (Fig. 2c III-VI). Both actin and DNA staining allowed a secondary visual validation of pre-mitotic cell rounding. As we recently described using comparable experiments[5], β1-integrin is shown to retain a similar spatial pattern similar to zyxin when the cells were spread on the substrate, in their pre-division shape (Fig. 2d). We observed that, while zyxin retracts during cell rounding (Supplementary Fig. 5b), β1-integrin remains in its original position (Fig. 2d and Supplementary Movie 1), helping guide daughter cell migration[5].

**Unsupervised live-to-fixed microscopy**. We then carried an experiment where the decision to activate a fluidics procedure is computationally driven, by continuously observing a sample and acting upon the identifying of a desired morphological cue. Here, automated detection of mitotic cell rounding was used to trigger live-to-fixed cell imaging protocols. For this, we used BeanShell scripting in μManager to programmatically define a workflow which integrates live-cell imaging, event detection, NanoJ-Fluidics fixation and staining followed by post-staining z-stack

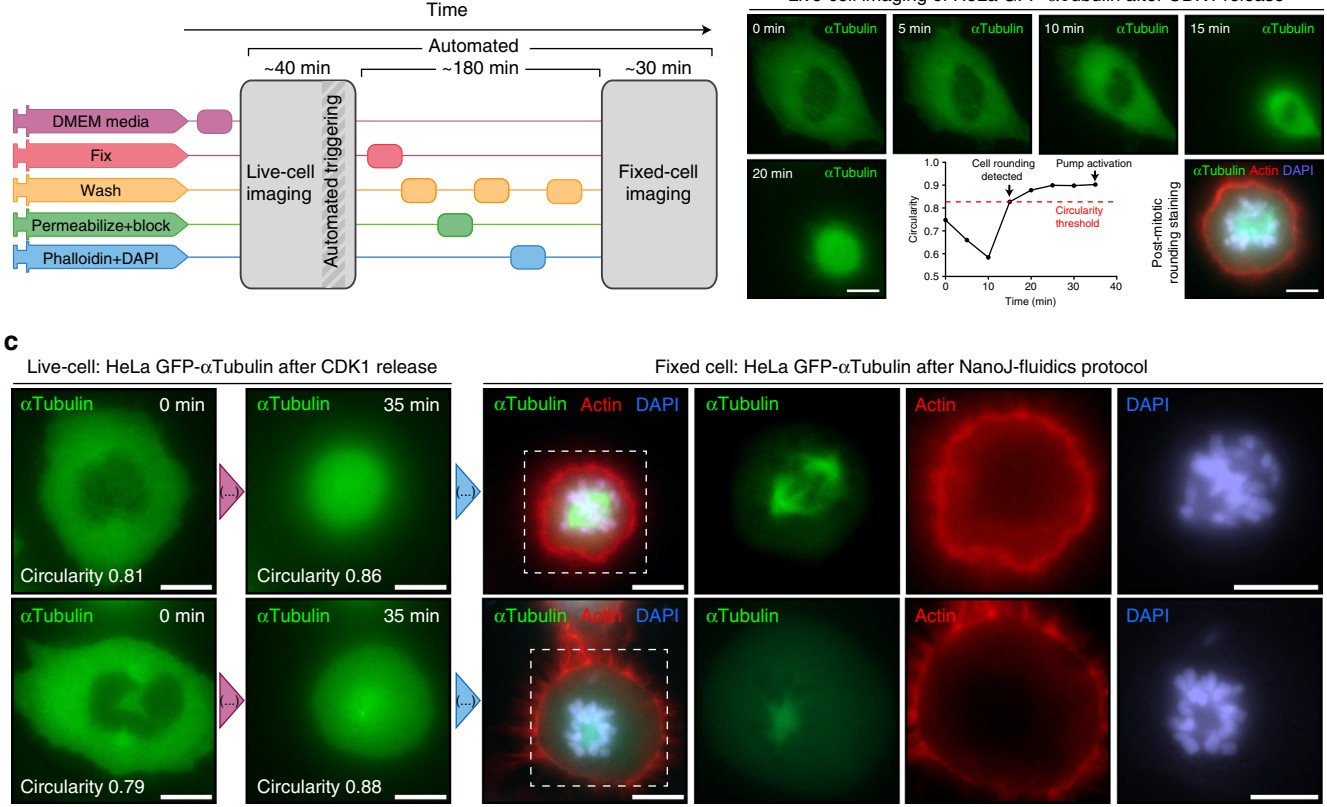

**Fig. 3** Unsupervised live-to-fixed microscopy triggered by mitotic rounding. **a** NanoJ-Fluidics workflow for the automated mitotic-rounding-driven protocol. **b** Stills of a HeLa cell expressing mEGFP-α-Tubulin (green) imaged live every 5 min (Supplementary Movie 2) during mitotic cell rounding, corresponding circularity analysis (graph) and overlay of the same cell after the NanoJ-Fluidics protocol (stained for actin—red, phalloidin-AF647 - and DNA—blue, DAPI). Rounding of cell shown was detected at 15 min. Pumps were activated at 35 min, 15 min after 25% of cells reached circularity threshold (Supplementary Note 5). Scale bar, 10 μm. **c** Two fields-of-view from the same experiment as in panel (**b**) where mitotic rounding occurred and corresponding single-plane (full z-stack shown in Supplementary Movie 2) of fixed-cell imaging of post NanoJ-Fluidics protocol, overlays and single images coloured as in panel (**b**). Scale bar, 10 μm

fixed-cell multicolour imaging. Given that the user can now be absent from the microscope, we also created an email-notification system that informs the user when the workflow advances to critical stages of the protocol.

We synchronised HeLa H2B-mCherry/mEGFP-α-Tubulin-expressing cells using CDK1 inhibition, as before. Using NanoJ-Fluidics, we first removed the inhibitor to release cells into cell cycle. We then imaged mEGFP-α-Tubulin every 5 min to observe mitotic cell rounding (Fig. 3 and Supplementary Movie 2). Here, we imaged 45 different regions of interest in the sample over time. When a sufficient fraction of observed cells was detected as rounded (25% of cells reaching a circularity >0.82, Fig. 3b), a 15 min counter was started leading to the activation of the automated in-situ fixation and staining of actin and DNA (phalloidin-AF647 and DAPI) (Fig. 3a). This 15 min delay ensures that the fixation protocol reflects cells already undergoing mitosis.

In Fig. 3b, the rounding for an individual cell was detected at 15 min. Here, the pumps were activated at 35 min, 15 min after 25% of detected cells reached the circularity threshold. After completion of the NanoJ-Fluidics protocol, fixed-cell imaging was performed to obtain the full 3D structure of the mitotic cell (Fig. 3b, c). The automation of the imaging and event detection allows seamless high-throughput imaging and simplifies the implementation of complex and laborious protocols on the microscope.

**Live-to-fixed super-resolution microscopy.** Next, we used NanoJ-Fluidics for a range of advanced super-resolution imaging protocols, demonstrating its performance and versatility.λLive-cell sub-diffraction microscopy is well established[10–12]. However, the near-molecular scale resolution obtained by single-molecule localisation microscopy (SMLM) is generally limited to fixed-cell imaging, due to the need for toxic buffers, reagents or illumination required for STORM[13] and DNA-PAINT[14]. Here we address this issue by using NanoJ-Fluidics to combine live and fixed-cell observation on the same set of cells, sequentially imaging the cellular actin dynamics and nanoscale architecture (Fig. 4a). Thanks to its cell-friendly low-illumination requirements, the SRRF approach was chosen to super-resolve actin labelled with a utrophin domain (UtrCH-GFP) probe (Fig. 4b). In order to quantify the resolution achieved here, we estimated the minimum (best) and mean resolutions within the cell (in-cell resolution, see Supplementary Note 5), using NanoJ-SQUIRREL[15] and confirmed that SRRF achieved an improved resolution over the standard HILO image (minimum in-cell resolution of 172 nm and 231 nm, respectively, Supplementary Fig. 5a), resolving the assembly and disassembly of actin bundles during cell-shape changes (Fig. 5b and Supplementary Movie 3). Then, we used NanoJ-Fluidics to perform automated fixation, washing, permeabilisation, phalloidin-AF647 staining—a sequence lasting 100 min (Fig. 4a)—and finally perfusion of a photoswitching buffer for STORM acquisition (Fig. 4c). STORM provided a greatly

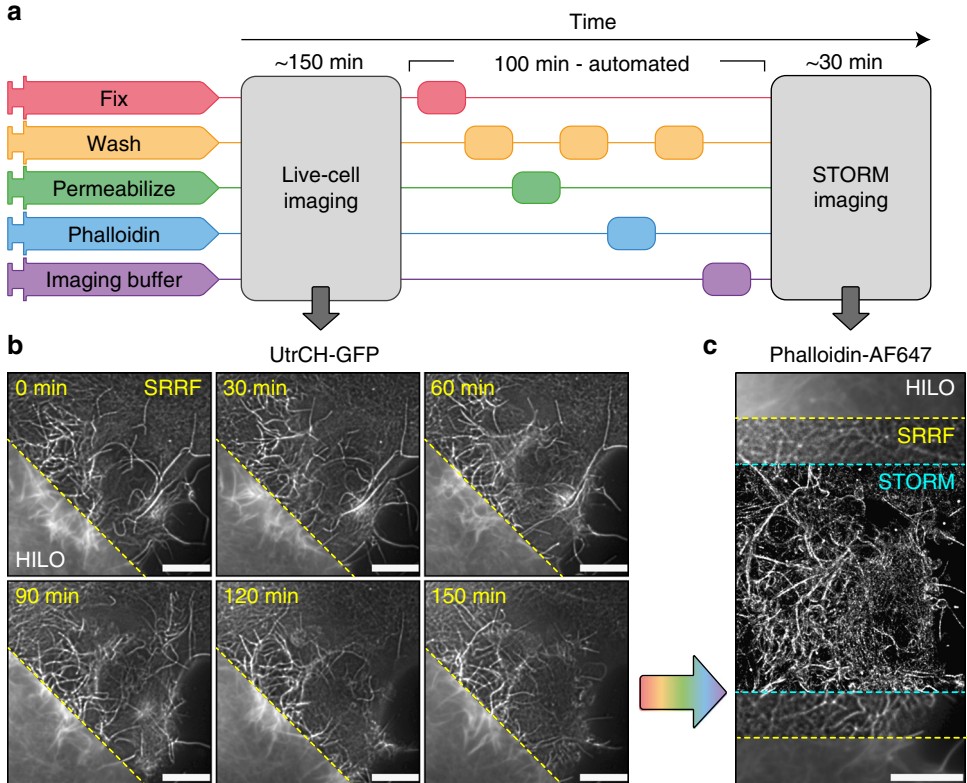

**Fig. 4** Super-resolution live-to-fixed cell imaging of actin. **a** NanoJ-Fluidics workflow used for live-to-fixed super-resolution imaging. **b** HILO and SRRF microscopy images of a COS7 cell expressing UtrCH-GFP imaged every 10 min for 150 min (zoomed in region is shown at 30 min intervals, Supplementary Movie 3 shows extended timelapse and field-of-view). **c** HILO and SRRF microscopy images of UtrCH-GFP at $t = 150$ min and the corresponding STORM image after fixation and staining with phalloidin-AF647. Scale bars are 10 μm

enhanced resolution for the actin organisation (minimum in-cell resolution of 43 nm, Supplementary Fig. 5b). Interestingly, these correlated observations also make it possible to evaluate if there are structural changes of the actin network caused by the fixation process (Supplementary Note 5 and Supplementary Fig. 6).

**Multiplexed STORM and DNA-PAINT super-resolution microscopy**. Obtaining a super-resolution, high-quality multi-colour SMLM image has long been a challenge as imaging condition are difficult to optimise for several labels simultaneously[16]. Recent DNA-PAINT modalities can reach more than the typical 2–3 channels by using antibodies coupled to orthogonal DNA strands and sequential labelling combined with imaging sequences of PAINT imagers[3,17]. The NanoJ-Fluidics automated fluid handling is ideally suited here and makes such multiplexing simple and robust. We demonstrate this with a 5-channel sequential STORM and DNA-PAINT acquisition scheme. Fixed cells were labelled using primary antibodies targeting mitochondria, vimentin, microtubules and clathrin followed by DNA-coupled secondary antibodies (Cy3B labelled imager strands for vimentin and clathrin, ATTO655 labelled imager strands for mitochondria and microtubules), as well as fluorescent phalloidin-ATTO488 to label actin. NanoJ-Fluidics enabled the exchange of imaging buffer for STORM imaging of actin, followed by two rounds of washing and labelling for 2-colour DNA-PAINT of the other targets (Fig. 5a and Supplementary Movie 4). The first round imaged mitochondria and vimentin whereas the second round imaged clathrin and tubulin. Figure 5b, c, respectively, show the resulting 5-colour image and the individual channels, along with zooms on cellular regions highlighting the

high resolution obtained by this combined scheme (~67 nm minimum in-cell resolution, except actin which has a 97 nm minimum, see Supplementary Note 4 and Supplementary Fig. 7).

## Discussion
We introduce NanoJ-Fluidics (Fig. 1) and demonstrate its applicability in multiple experimental con-texts: event-driven sample treatment (Figs. 2 and 3), in-situ correlative live-to-fixed super-resolution imaging (Fig. 4) and multimodal multilabel super-resolution imaging (Fig. 5). The software modules of NanoJ-Fluidics are part of our NanoJ microscopy toolbox[18] developed in ImageJ/Fiji[19,20]. The reliability provided by the computer-controlled fluidic experiments (Supplementary Note 2 and 3) overcomes the issues of repeatability and low throughput of traditional, non-automated approaches. We also show that NanoJ-Fluidics can be integrated with other automated systems by performing sample treatment triggered by a specific biological cues (mitotic cell rounding) through the integration of NanoJ-Fluidics with the image acquisition (Figs. 2 and 3 and Supplementary Movie 1 and 2). NanoJ-Fluidics further enables to correlate in-situ the dynamic information from live-cell imaging with the nanoscale structural information from fixed-cell SMLM (Fig. 4), without the laborious relocation of multiple cells or manual immunolabelling steps. These experiments showcase the broad scope of NanoJ-Fluidics in increasingly complex experimental settings and highlights its potential in the context of recently described high-throughput, high-content approaches[21,22]. This paves the way towards unsupervised, high-content, event-driven correlative live-to-fixed imaging or

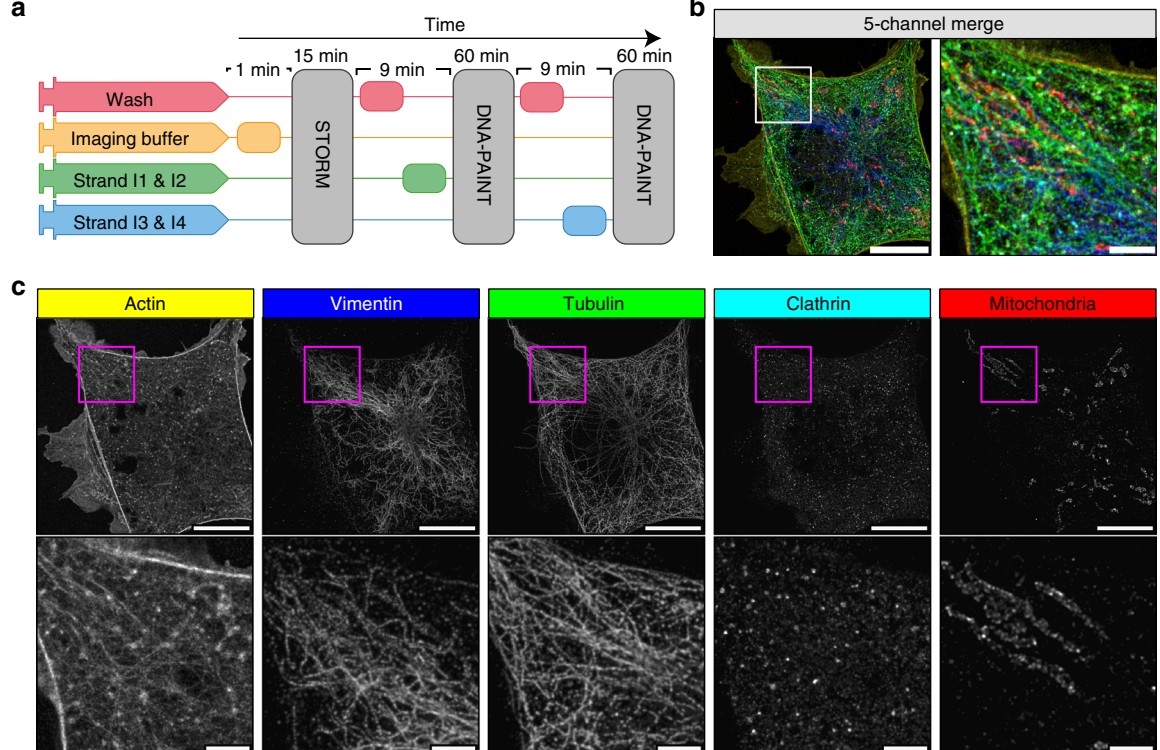

**Fig. 5** Automated DNA-PAINT and STORM imaging. **a** NanoJ-Fluidics workflow used for STORM and DNA-PAINT imaging. **b** Left, full view showing 5-channel merge of STORM and DNA-PAINT with actin (yellow, Phalloidin-ATTO488), vimentin (blue, Cy3B DNA-PAINT imager strand), β-tubulin (green, ATTO655 DNA-PAINT imager strand), clathrin (cyan, Cy3B DNA-PAINT imager strand) and mitochondria (red, ATTO655 DNA-PAINT imager strand). Right, zoom of the boxed area. **c** Single-channel image of each imaged target (top), with insets (bottom) showing a zoom of the boxed area. A movie corresponding to this experiment is available as Supplementary Movie 4. Scale bar corresponds to 10 μm for full images and 2 μm for zooms

sample treatment compatible with highly flexible experimental workflows of imaging and fluidics exchange.

Besides this potential for complex biological interrogation, the simplicity and robustness of NanoJ-Fluidics facilitates experimental optimisation (Supplementary Note 5)[15]. For example, to reach optimal SMLM images it is often necessary to fine-tune fixation, permeabilisation, blocking, antibodies concentration and imaging buffer composition[23]. However, tuning each factor and their combinations is a daunting task. The NanoJ-Fluidics automation and its integration with imaging allows for a more comprehensive and reliable exploration of these settings. Also, SMLM microscopy approaches tend to be restricted to a couple of colours due to fluorophore photophysics requiring specific imaging buffers. NanoJ-Fluidics provides significant advantages in this context, by allowing, for instance, the easy automation of STORM[16] with sequential DNA-PAINT[3] protocols (Fig. 5). It also holds the potential to perform other sequential labelling protocols based on bleaching or antibody removal[6,24,25].

NanoJ-Fluidics is easy (and fun) to set up and use (Supplementary Note 1 and Wiki: https://github.com/HenriquesLab/NanoJ-Fluidics/wiki). It is also low-cost and composed by parts easy to acquire (Supplementary Table 1). Its modular design is highly adaptable: for example, pump modules can be detached to be kept at different temperatures (Supplementary Fig. 1). NanoJ-Fluidics design has several advantages compared to microfluidics chips: glass-bottom chambers are easier to prepare: air bubbles formed in the tubing are easily dissipated before damaging the sample; ambient pressure allows for a higher tolerance to flow-rate fluctuations. Notwithstanding, microfluidics devices have the general advantage of requiring smaller volumes of reagents. Finally, NanoJ-Fluidics can be used with other third-party,

commercially available pumps, as the open-source software provides an Application Programming Interface (API) and plugin framework for control within the same software interface (Supplementary Fig. 3).

In this work, we have chosen to focus on NanoJ-Fluidics applications to live-cell imaging and SMLM combinations. However, our fluidic exchange system can be used in any experiment that would benefit from an automated exchange of the sample media. As highlighted by our unsupervised detection of mitotic cell rounding and sample treatment (Fig. 3 and Supplementary Movie 2), our μManager plugin therefore allows for NanoJ-Fluidics to be fully integrated into microscopy acquisition software, enabling a seamless combination between the imaging and fluid exchange protocol. In conclusion, NanoJ-Fluidics makes high-content multimodal microscopy experiments tractable and available to a large audience of researchers, while improving reliability, optimisation and repeatability of imaging protocols.

## Methods

**Cell lines**. COS7 (ATCC CRL-1651) and HeLa H2B-mCherry/mEGFP-α-Tubulin (HCTG, kind gift from Daniel W. Gerlich) stable cell line[26,27] cells were cultured in phenol-red free DMEM (Gibco) supplemented with 2 mM GlutaMAX (Gibco), 50 U/ml penicillin, 50 μg/ml streptomycin (Penstrep, Gibco) and 10% foetal bovine serum (FBS; Gibco). hTERT-RPE1 cells stably expressing zyxin-GFP[5] were cultured in DMEM F-12 GlutaMAX (Gibco), with 10% FBS, 3.4% sodium bicarbonate (Gibco), 1% Penstrep. All cells were grown at 37 °C in a 5% CO₂ humidified incubator. Cell lines have not been authenticated.

**Plasmids**. The plasmid expressing the calponin homology domain of utrophin fused to GFP (UtrCH-GFP) was a gift from William Bement[28] (Addgene plasmid #26737).

**Antibody conjugation**. Secondary antibodies were labelled with the corresponding DNA strands using the following protocol[29]: secondary antibodies were concentrated via Amicon 100 kDa spin filters to 2–6 mg/ml. A volume of 50–100 μg of antibody was labelled using a Maleimide-Peg2-succinimidyl ester (stocks of 10 mg/ml in DMF) for 90 min at 40× molar excess at 4 °C on a shaker. After the 90 min incubation, unreacted crosslinker was removed via a Zeba spin column. Thiolated DNA was reduced using DTT for 2 h at room temperature. DTT was separated from the reduced DNA via a Nap5 column and fractions containing DNA were concentrated via 3 kDa Amicon spin filters. The reduced DNA was then added to the antibody bearing a functional maleimide group in 25× molar excess and incubated overnight at 4 °C on a shaker in the dark. Antibody-DNA constructs were finally purified via 100 kDa Amicon spin filters. DNA-PAINT labelling:

For Mitochondria: Goat anti-Mouse (ThermoFisher A28174,) with I1 (docking: 5′-TTATACATCTA-3′; imager: 5′-CTAGATGTAT-ATTO655-3′);
For Vimentin: Goat anti-chicken (Abcam ab7113) with I2 (docking: 5′-TTAATTGAGTA-3′; imager: 5′- GTACTCAATT-Cy3B-3′);
For Clathrin: Goat anti-Rabbit (ThermoFisher A27033). with I3 (docking: 5′-TTTCTTCATTA-3′; imager: 5′- GTAATGAAGA-Cy3B-3′);
For alpha-tubulin: Donkey anti-Rat (ThermoFisher A18747) with I4 (docking: 5′-TTTATTAAGCT-3′; imager: 5′-CAGCTTAATA-ATTO655-3′).

Sequences for DNA-PAINT strands were obtained from[3]. Both thiolated and fluorophore conjugated DNA strands were obtained from Metabion.

**NanoJ-Fluidics framework**. We provide the detailed instructions to easily build and use the system in a regular biology lab, as well as the software enabling its control and automation (Supplementary Note 1).

**NanoJ-Fluidics Lego syringe pump calibration**. NanoJ-Fluidics Lego syringe pump calibration was performed by injecting predefined volumes into a container sitting inside an analytical scale (Sartorius Cubis®, 0.1 mg ± 0.0003 mg). The procedure was repeated using different syringe pumps, and syringe sizes (see Supplementary Note 2 and 3 for discussion). The dispensed volume was weighed and an ordinary one-way ANOVA was used (GraphPad Prism version 7.00 for Mac, GraphPad Software, La Jolla California USA) to estimate the statistical difference between the different conditions used in Supplementary Fig. 4 (using a confidence interval of 99%). For the Overflow loading mode, a volume of 100 μL was measured with a Gilson P100 micropipette (by weighing it in an analytical scale, Sartorius Cubis®, 0.1 mg ± 0.0003 mg). The weighed volume was inserted into the tubing and then dispensed using the pump by infusing a 500 μL volume. By flushing out a larger volume than injected, this method ensures that the entire volume is dispensed.

**Live-to-fixed super-resolution imaging**. The NanoJ-Fluidics syringe pump array was installed on a Nikon N-STORM microscope equipped with 405, 488, 561 and 647 nm lasers (20, 50, 50 and 125 mW at the optical fibre output). One individual syringe pump module containing the fixative was kept within the incubator of the microscope at 37 °C. All steps after cell transfection were performed on the microscope, using NanoJ-Fluidics. COS7 cells (kind gift from M. Marsh) were seeded on ultraclean[30] 25 mm diameter thickness 1.5 H coverslips (Marienfeld) at a density of 0.3–0.9 × 10^5 cells/cm². One day after splitting, cells were transfected with a plasmid encoding the calponin homology domain of utrophin fused to GFP (UtrCH-GFP) using Lipofectamin 2000 (Thermo Fisher Scientific) according to the manufacturer's recommendations. Cells were imaged 1–2 days post transfection in culture medium using an Attofluor cell chamber (ThermoFisher), covered with the lid of a 35 mm dish (ThermoFisher), that was kept in place using black non-reflective aluminium tape (T205-1.0 AT205, THORLABs).
Cells were fixed at 37 °C for 15 min with 4% paraformaldehyde in the cytoskeleton-preserving buffer "PIPES-EGTA-Magnesium" (PEM: 80 mM PIPES pH 6.8, 5 mM EGTA, 2 mM MgCl2)[31]. After fixation cells were permeabilised (PEM with 0.25% Triton-X) for 20 min, blocked with blocking buffer (5% Bovine Serum Albumin (BSA) in PEM) for 30 min, and stained with Phalloidin-AF647 (Molecular Probes, 4 units/mL) for 30 min.
Laser-illumination Highly Inclined and Laminated Optical sheet (HILO) imaging of UtrCH-GFP in live COS7 cells was performed at 37 °C and 5% CO₂ on a Nikon N-STORM microscope. A 100x TIRF objective (Plan-APOCHROMAT 100 × /1.49 Oil, Nikon) with additional ×1.5 magnification was used to collect fluorescence onto an EMCCD camera (iXon Ultra 897, Andor), yielding a pixel size of 107 nm. For timelapse imaging, 100 raw frames (33 ms exposure) were acquired once every 10 min (with the illumination shutter closed between acquisitions) for 150 min with 488 nm laser illumination at 4% of maximum output. STORM HILO imaging of Alexa Fluor 647-phalloidin in fixed cells was performed on the same system. About 50,000 frames were acquired with 33 ms exposure and 642 nm laser illumination at maximum output power with 405 nm pumping when required (0.5–1% of maximum output when the blinking density was bellow 1 particle/μm²). STORM imaging was performed in GLOX buffer (150 mM Tris, pH 8, 1% glycerol, 1% glucose, 10 mM NaCl, 1% β-mercaptoethanol, 0.5 mg/ml glucose oxidase, 40 μg/ml catalase) supplemented with Phalloidin-Alexa Fluor 647 (1 U/mL).

**Multiplexed DNA-PAINT and STORM super-resolution microscopy**. COS7 cells (obtained from ATCC) were seeded on 18 mm, 1.5 H glass coverslips (Menzel-Gläser). About 24 h after seeding, they were fixed using 4% PFA, 4% sucrose in PEM at 37 °C[31]. After blocking in phosphate buffer with 0.022% gelatin, 0.1% Triton-X100 for 1.5 h, cells were incubated with primary antibodies overnight at 4 °C: mouse monoclonal anti-TOM20 (BD Bioscience 612278), rabbit polyclonal anti-clathrin heavy chain (abcam ab21679), chicken polyclonal anti-vimentin (BioLegend 919101) and rat anti-alpha-tubulin (mix of clone YL1/2 abcam 6160 and clone YOL1/34 Millipore CBL270). After rinses, they were incubated with Exchange-PAINT secondary antibodies coupled to DNA sequences: goat anti-mouse I1, goat anti-chicken I2, goat anti-rabbit I3 and donkey anti-rat I4 for 1.5 h at RT (see Ab conjugation section for antibody and sequence details). After rinses, they were incubated in phalloidin-Atto488 (Sigma) at 12.5 μM for 90 min at RT and imaged within a few days[32]. For STORM/PAINT imaging, the NanoJ-Fluidics array was installed on an N-STORM microscope (Nikon) equipped with 405, 488, 561 and 647 nm lasers (25, 80, 80 and 125 mW at the optical fibre output). First, a STORM image of phalloidin-ATTO488 was acquired in buffer C (PBS 0.1 M pH7.2, 500 mM NaCl) by taking 30,000 frames at 30 ms/frame at 50% power of the 488 nm laser. After injection of the I1-ATTO655 (0.25 nM) and I2-CY3B (2 nM) imagers in buffer C, 60,000 frames were acquired by sequentially interleaving the two channels (60% power of the 647 nm laser and 30% power of the 561 nm laser) to image TOM20 and vimentin in a single acquisition, therefore providing 30,000 frames for each channel. After three rinses with buffer C, I3-Cy3B (1 nM) and I4-ATTO655 (0.5 nM) were injected in buffer C, and 60,000 frames were similarly acquired via frame interleaving (30% power of the 561 nm laser and 60% power of the 647 nm laser, 30,000 frames of each channel) to image clathrin and microtubules, respectively.

**Event detection and live-to-fix imaging**. hTERT-RPE1 cells stably expressing zyxin-GFP were incubated with 9 mM RO-3306 (Enzo Life Sciences ALX-270-463) to inhibit CDK1 activity for 15–20 h. Inhibition was released by replacing drug containing media by fresh media at the microscope immediately before imaging. Cells were imaged using a Nikon Eclipse Ti microscope (Nikon) equipped with a Neo-Zyla sCMOS camera (Andor), LED illumination (CoolLED) and a ×60 objective (Plan Apo 60 × /1.4 Oil, Nikon). Images were acquired every 3 min until enough cells had undergone mitotic rounding. At that point, determined by visual inspection, 16% warmed PFA (from a Lego syringe pump placed inside the incubation chamber) was added to cells in media to a final concentration of 4%, and incubated at room temperature for 20 min. They were then washed three times and 0.2% Triton was added for 5 min. 5% BSA in 1X PBS was used to block for 30 min at room temperature, before activated β1 Integrin (Abcam ab30394) primary antibody was added. After incubation and washing, Phalloidin-TRITC (Sigma-Aldrich) and anti-mouse AF647 antibody (Invitrogen) were added. All of these steps were performed automatically using the NanoJ-Fluidics platform, except the identification of the time point when to fix the cells.

**SMLM and SRRF image reconstruction**. For Fig. 4 images were reconstructed using NanoJ-SRRF[12] (magnification: 4, temporal analysis settings: TRPPM for live-cell data and TRM for fixed-cell data). Drift was estimated using the inbuilt function in NanoJ-SRRF and correction applied during SRRF analysis. For Fig. 5 localisations were detected using the N-STORM software (Nikon), and exported as a text file before being filtered (number of photons between 700 and 50,000; number of detections (after linking across frames) <50 frames) and rendered using ThunderSTORM[33]. Additionally, a 32 nm radius Gaussian blur (ImageJ/Gaussian blur) and a Gamma correction (ImageJ/Gamma, varying between 0.6 and 1.5) were applied. Chromatic aberration between the red (561 nm) and far-red (647 nm) channels were corrected within the N-STORM software using polynomial warping and remaining translational drift between acquisition passes were aligned manually on high-resolution reconstructions.
The image resolution was estimated by calculating the Fourier Ring Correlation (FRC)[34] with the typical 1/7 threshold using NanoJ-SQUIRREL plugin and reconstructing the original dataset separated into two different stacks composed of odd or even images[15]. NanoJ-SRRF, NanoJ-SQUIRREL and ThunderSTORM are available in Fiji[20].

**Unsupervised NanoJ-Fluidics and integration with μManager**. HeLa HRTG were plated on 35 mm iBidi dishes and incubated with Cdk1 inhibitor (RO-3306, 5 nM) for 15–20 h before imaging. Immediately prior to imaging, cells were washed with fresh medium using NanoJ-Fluidics and live-cell imaging was started. For each experiment, 30–50 fields-of-view were selected and imaged every 5 min in the GFP and mCherry channels. Imaging was performed at 37 °C and 5% CO₂ on a Nikon Eclipse Ti with a CFI Apo TIRF 60X Oil (1.49 NA), LED illumination (CoolLED P-4000) and Andor Zyla 4.2 sCMOS camera (100 ms exposure time and 20% illumination for each channel). A BeanShell script (available on our GitHub page) was designed to run the acquisition and image analysis in parallel. For each field-of-view, cells were segmented by Otsu thresholding followed by morphological opening. The circularity of each cell was calculated and if the average circularity of a field-of-view exceeded a user-defined circularity threshold (here set to 0.82) the field-of-view was declared as triggered. The circularity was estimated via

the common shape factors measurement of ImageJ,

$$C = \frac{4\pi A}{P^2} \qquad (1)$$

where $C$ is the circularity estimator, $A$ is the area and $P$ is the perimeter.

Once a user-set fraction of fields-of-view triggered (here 25%), the 'cell rounding event' was activated. At that stage, NanoJ-Fluidics fixation and labelling protocol were initiated (a small delay was added to visualise the evolution of the rounded cell further, here 15 min). The NanoJ-Fluidics sequence was set as follows:

Fixation with 4% PFA in PEM buffer for 15 min at 37 °C.
Wash with 1X PBS for 10 min, (×3)
Permeabilisation and blocking in 1X PBS with 5% BSA and 0.05% Triton-X for 30 min.
Staining with Phalloidin-AlexaFluor647 (4 units/mL) and DAPI (1 µg/mL) for 1 h.
Wash with 1X PBS for 10 min, (×2)

Once the fluidics steps were executed, the post-fixation/staining imaging was then automatically performed taking a 3D z-stack of 30–40 planes (spaced by 0.5 µm) in DAPI, GFP, mCherry and AlexaFluor647 channels (200 ms exposure times and 20% illumination for each channel).

**Reporting summary**. Further information on experimental design is available in the Nature Research Reporting Summary linked to this article.

**Code availability**. NanoJ-Fluidics follows open-source software and hardware standards, it is part of the NanoJ project[12,15]. The steps to assemble a complete functioning system are described in https://github.com/HenriquesLab/NanoJ-Fluidics.[35]

## Data availability
The data that support the findings of this study are available from the corresponding author upon reasonable request.

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

## Acknowledgements
We thank Prof. Ralf Jungmann at Max Planck Institute of Biochemistry Munich for reagents and advice. This work was funded by grants from the UK Biotechnology and Biological Sciences Research Council (BB/M022374/1; BB/P027431/1; BB/R000697/1; BB/R021805/1) (R.H., P.M.P. and R.F.L.), the UK Medical Research Council (MR/K015826/1) (R.H.), the Wellcome Trust (203276/Z/16/Z) (S.C., B.B and R.H.) and the Centre National de la Recherche Scientifique (CNRS ATIP-AVENIR program AO2016) (C.L.). P.A. was supported by a PhD fellowship from the UK's Biotechnology and Biological Sciences Research Council. C.L.D. was supported by PhD funding from the Medical Research Council, UK (1214605). Research by B.B. was supported by UCL, Cancer Research UK (C1529/A17343), and MRC (MC_CF12266).

## Author contributions
P.A. and R.H. devised the hardware and wrote the software. P.A., P.M.P., R.F.L., C.L. and R.H. planned experiments. Experimental data sets were acquired by P.M.P. (Fig. 4), G.C, F.B.-R. and C.L. (Fig. 5), P.M.P. (Supplementary Figure 4), P.M.P. and R.F.L. (Fig. 3), P.A. and C.L.D. (Fig. 2). Data was analysed by P.A., P.M.P., S.C and R.F.L. while G.C., B.B., C.L. and R.H. provided research advice. The paper was written by P.A. P.M.P., R.F.L., C.L. and R.H. with editing contributions of all the authors.

## Additional information

**Competing interests:** The authors declare no competing interests.

