## [Peer Review File · Nature Communications]

Reviewers' comments:

Reviewer #1 (Remarks to the Author):

The authors present an interesting low-budget approach to fluid exchange for microscopic imaging. From a scientific perspective there is very little that is novel about what is described. Should accuracy be sufficient, however, the approach could prove very attractive as a tool as it is likely to represent a considerable saving over existing commercial alternatives - particularly those with multiple independent perfusion channels. In addition to the hardware, there is what appears to be a useful open-source control software offering. I think the paper is neat, but the scientific novelty (or lack thereof) makes me uncertain if it is quite Nature Communications level.

I have one major concern, which is accuracy/reproducibility. Whether or not this is useful to the community as a whole will depend critically on the accuracy of the system and I feel that this has not been adequately investigated. As it stands, the accuracy measurements which have been made (indicating a 1 std. dev. accuracy of $\pm \sim 5\%$) would suggest that the system is barely accurate enough to be useful. $\pm 5\%$ may sound OK, but because errors are cumulative a 5% error between input and output rates would be enough to either overflow or run the bath dry within 20 media changes. Given that you'd want around 4-5 media changes for each rinsing step this is likely to represent a critical limitation on protocol length.

The given accuracy measurements are not described in detail.

- Was this a single syringe (for each volume) and repeated extrusions, or was it separate syringes?
- Was it a single pump? How does extrusion rate vary between pumps?
- The motors used are direct drive and there is no encoder / feedback meaning that the amount of motion will vary depending on load [a stepper motor or continuous rotation servo would have been a much better choice here, and would not have added much to the overall cost]. Does extrusion volume depend on how much is left in the syringe? How about reagent viscosity / surface tension / liquid head? Is extruded volume linear in extrusion time?

I would like to see the reproducibility quantified:

- in a single syringe over the full volume
- across syringes
- across pumps
- across different speeds and activation durations

The Arduino code linked on Github is such that motor control and IO are performed in a sequential loop. This will put some constraints on timing accuracy and hence minimum extrusion volume / extrusion accuracy at small volumes. I would imagine that the accuracy at small volumes/times would be considerably worse than with a 10s duration. Whilst this is not a critical problem, the authors should comment on this and state where the practical limits lie.

It would additionally be nice (and this is of relatively minor importance) if a full BOM with indicative costs were provided so as to give a quick indication of how the cost compares to commercial options.

Reviewer #2 (Remarks to the Author):

General points:

This paper is a presentation of an open source hardware design and software for automating multi-channel fluid exchange (sample media and imaging reagents) in fluorescence imaging. It claims to

make high-content multimodal experiments microscopy tractable, available and more repeatable. Live-to-fixed cell correlative imaging (using dSTORM after fixation), multiplexed super-resolution imaging (dSTORM and exchange-DNA-PAINT) and automatic, event-triggered fixation are demonstrated (although the latter is only provided in a supplementary note). The paper makes the point that published microfluidic solutions or manual fluid exchange protocols have been possible, but does promise easier (compared with microfluidics) or more automatic and reliable (compared with manual exchange) setup and execution of experiments. Together with the open source design, setup and software details, I expect this to be of interest to developers of microscope hardware and protocols, and to those who require such media exchanges in their imaging experiments. It is a design that I expect laboratories will consider testing or developing further, and will provide a clear advantage over manual methods (e.g. those similar to previous exchange-DNA-PAINT: Jungmann, Nat. Methods (2014), as cited) when automation of the fluid exchange is a requirement of an experiment. When automation or remote control of fluidics is not strictly necessary, this system will have less impact. It is possible that important experiments necessitating fully automated or remote controlled media exchange will increase in number as a result of the presentation and availability of this design.

The automatic, event-triggered fixation demonstrated in a supplementary note seems the clearest example of where NanoJ-Fluidics would come into its own, if mitotic rounding and subsequent fixation are detected and triggered without user input at that point. Is that done without user input? If not, can that be done? The potential to put a pump inside a sealed incubator (Fig S1) is another example of this requirement for and usefulness of automation, or remote control. In other examples, automation is convenient, but perhaps less necessary, and manual use of the same syringes and tubes may suffice, although it would require more time spent at the instrument. There will be other potential applications where the automation or remote control is necessary. These may be a better demonstration of the usefulness of the NanoJ-Fluidics concept in the main paper, or other advantages not specifically discussed could perhaps be presented.

The definitions of resolution (discussed below) need to be made consistent. The work appears to be clearly reproducible in general, with a few extra details required (requested below).

Specific points

1. Introduction

'Carrying elaborate protocols, interlacing imaging and manual media replacement, critically hampers their reproducibility...': This sentence is difficult to read clearly.

2. Construction / Supplementary note 1

While LEGO is easily available, the instructions the reader is directed to on the internet indicate that not all parts are available from the same source, and sourcing the Lego Technic components may be less reliable and more complicated, from second-hand sellers. A statement or list of how many sources (vendors) of LEGO components were required, or would currently be required to build the system of 4 or 8 pumps would be informative to the potential builder. Alternatively or additionally, 3D printing models could be provided that would allow builders to acquire the construction parts that way.

'polydimethylsiloxane' should not be capitalized.

3. Live-to-fixed super-resolution microscopy

The authors use SRRF processing to improve the resolution of their live-cell images, acquired with HILO illumination. However, the resolution comparison is confusing, since what is stated for SRRF appears to be the minimum resolution for the single region of the image with the smallest resolution over all regions and timepoints, whereas approximate HILO resolutions are provided at each timepoint. Later on, the dSTORM and DNA-PAINT images are characterised by 'minimum resolution', presumably as an indication of what is achievable using this protocol. Perhaps this

'minimum resolution' would be fairer terminology for the live-cell images, and usable for both SRRF and HILO, as well as the dSTORM image of Fig. 2. In this case, there is no need to give these figures as an approximation. If an indication of the average image resolution is desired, that would need defining differently. If the comparison between unprocessed and SRRF images needs to be made here, perhaps similar FRC maps for the unprocessed HILO image sequences could be provided as well.

4. Multiplexed STORM and DNA-PAINT

The definition of 'resolution' is clearer here, but can still be tightened by removing the approximation symbols and stating '66–68 nm minimum resolution, except for actin... 97 nm...', seeing as this seems to be well-defined. Please confirm that these regions of minimum resolution include parts of the cell – especially for the actin image. If not, what is the minimum resolution within the cell image?

Proof of the ability to multiplex super-resolution imaging is given here, but discussion of the resolutions achieved would be beneficial somewhere, since higher resolutions should be possible using NanoJ-Fluidics with alternative protocols. Probably, practical considerations will have led to the choice of ATTO488 for the actin image, which may have resulted in worse performance than the dSTORM image of actin in Fig. 2, for example. It would be good to explain this difference, and the reasons for this choice. In other images, the use of both primary and secondary antibodies will have decreased resolution compared to what is possible, for example with DNA-labelled primary antibodies (e.g. Jungmann, Nat. Methods (2014), as cited). There may be other known factors at work which would be helpful to understand. Such a discussion may then allow readers to see greater potential in the system.

For reproducibility, can the authors provide more specific information about the filtering and rendering procedure applied in ThunderSTORM?

5. Fig S6 / Supplementary note 3

I did not find it clear what the last sentence is about (loss of intensity in the top left of Fig S6). Is there a particular part of Fig S6 that this refers to?

6. Event-driven fixation imaging (supplementary note 4)

This application is potentially a good illustration of the usefulness of the automated fluidics, and would add significance to the main text of the paper, particularly if the mitotic rounding was detected automatically and triggered fixation without user input at that point (see general points). The boxes referred to as insets look like they are simply boxes to label cells; there do not appear to be inset images.

7. Discussion

NanoJ-SQUIRREL is given twice as a reference for how NanoJ-Fluidics allows easier optimisation of experimental protocols. I think the cross-reference to Sup. Note 3, where NanoJ-SQUIRREL is used is enough, and about right, to point out the potential connection for this purpose.

Similar to my general point, it is stated here that NanoJ-Fluidics provides significant advantages for such combinations as dSTORM and exchange-DNA-PAINT; but these media exchanges are already 'allowed' by a selection of syringes without automation (e.g. similar to Jungmann, Nat. Methods (2014), as cited). It would be good to hear what particular advantages over manual use of the syringes are in view here.

With regards to the comparison with microfluidics, concerning reagent volumes, it would be beneficial to hear how much dSTORM buffer is required to pass through the tubing in the implementation presented here.

I recommend that 'robustness of the NanoJ-Fluidics' should read 'robustness of NanoJ-Fluidics', for

consistency with its treatment elsewhere.

'For example, optimal SMLM images requires fine-tuning the fixation...' needs rephrasing.

8. Methods

Live-to-fixed imaging:

For reproducibility, what power 405 nm pumping was used, when it was necessary?

Multiplexed super-resolution:

Is the 'PEM buffer' the same as the 'cytoskeleton-preserving buffer' of the previous section? If so, it would be more clearly named again as such, or named PEM buffer the first time.

For reproducibility, what is the 'alternating way' in which frames are acquired for two-colour DNA-PAINT (for both I1/I2 and I3/I4)? For clarity, what do the 60% and 30% refer to? Is it laser output power, or something else? Can the authors be more precise about the concentrations for the various imager strands?

'I3-Cy3B and I4-ATTO655 were in buffer C were injected' should read 'I3-Cy3B and I4-ATTO655 were injected in buffer C'.

Event detection and live-to-fixed imaging:

Here or in the supplementary note, more detail would be good about the recognition of the appropriate timepoint for fixation: how was the 'minimal area' (suppl. note 4) recognised as such?

'underwent' should read 'undergone'.

Dear Reviewers and Editor,

We thank the reviewers for the overall positive assessment of NanoJ-Fluidics. Your feedback has been essential in helping achieve a considerably improved manuscript. We have included a substantial amount of new characterisation data in the revised version addressing key points raised in the reviewers' comments. These are summarised in a point-by-point response below:

Reviewer #1 (Remarks to the Author):

The authors present an interesting low-budget approach to fluid exchange for microscopic imaging. From a scientific perspective there is very little that is novel about what is described. Should accuracy be sufficient, however, the approach could prove very attractive as a tool as it is likely to represent a considerable saving over existing commercial alternatives - particularly those with multiple independent perfusion channels.

We thank the reviewer for the feedback and positive remark. We completely agree that the manuscript would benefit from an extended characterisation of the NanoJ-Fluidics performance to give users a perspective of its benefits. We now added Sup. Note 3 and Fig. S4 which shows an in-depth analysis of precision, accuracy and repeatability. We also added Sup. Note 2 which discusses the workflows NanoJ-Fluidics was designed for and where it should be expected to achieve an excellent performance.

Regarding novelty, we believe that NanoJ-Fluidics brings a unique combination of 4 key advantages that makes the system innovative compared to alternatives:

- **Accessibility:** It's a considerably inexpensive (we've added a cost breakdown in Table S1) and easy-to-build system compared to commercial equivalents. We hope that the fact that there are already 7 independent laboratories publicly reproducing the system showcases this point (ref. 1-7 below).

- **Versatility:** Being open-source and based on LEGO means the system can be easily modified for different applications, for example some of the labs reproducing NanoJ-Fluidics are already making variations to the pump design (ref 7-8).

- **Ease-of-use:** The entire system uses off-the-shelf well-known components. The software interface is based on ImageJ/Fiji/ μ Manager (well-known and highly used software in biology research) and has a GUI (graphical user interface) designed to easily replicate complex protocols, such as those showcased in the paper.

- **Integration with acquisition:** the automation interface is directly compatible with μ Manager, allowing researchers to programmatically integrate the fluid exchange protocols with the acquisition itself.

It is our belief that these features will attract researchers to build and employ NanoJ-Fluidics in their own research. The platform has already raised considerable interest within the research community.

References:

- 1 - <https://twitter.com/alexcarisey/status/1024641256004173825> (A. Carisey, Columbia Univ. Med.)
- 2 - https://twitter.com/PhelpsLab_UF/status/1024648529925615616 (E. Phelps, Univ. Florida)
- 3 - https://twitter.com/V_Saggiomo/status/1024641218230345728 (V. Saggiomo, Wageningen Univ.)
- 4 - <https://twitter.com/FritzPreusser/status/1024998990356656128> (S. Preibisch, MDC Berlin)
- 5 - https://twitter.com/Chris_W_Ward/status/1024771453693120512 (C. Ward, Univ. Maryland)
- 6 - <https://twitter.com/NicholasMG/status/1001803661448654848> (D. Shepherd, Univ. Colorado)
- 7 - <https://twitter.com/LJIMicroCore/status/1025551246051012610> (Z. Mikulski, La Jolla Institute)
- 8 - https://twitter.com/V_Saggiomo/status/1010143406784417792 (V. Saggiomo, Wageningen Univ.)

... I think the paper is neat, but the scientific novelty (or lack thereof) makes me uncertain if it is quite Nature Communications level.

While we understand the reviewer's point of view, we believe that there is considerable merit to publications making established technologies widely available to an audience that otherwise would have limited access to them. There are good examples that can be found published on journals with high-standards, for instance QuickPALM (1) and OpenSPIM (2). Both these approaches are not the first of their kind, but have made considerable impact in their own fields by making SR analysis and SPIM imaging broadly accessible to the research community. We expect the interest and uptake already shown by NanoJ-Fluidics to be a good indication that it will follow a similar path.

References:

- 1 – Henriques et al., Nat. Meth., 2010, <https://doi.org/10.1038/nmeth0510-339>
- 2 – Pitrone et al., Nat. Meth., 2013, <https://doi.org/10.1038/nmeth.2507>

I have one major concern, which is accuracy/reproducibility. Whether or not this is useful to the community as a whole will depend critically on the accuracy of the system and I feel that this has not been adequately investigated. As it stands, the accuracy measurements which have been made (indicating a 1 std. dev. accuracy of +/- ~5%) would suggest that the system is barely accurate enough to be useful. +/-5% may sound OK, but because errors are cumulative a 5% error between input and output rates would be enough to either overflow or run the bath dry within 20 media changes. Given that you'd want around 4-5 media changes for each rinsing step this is likely to represent a critical limitation on protocol length.

This comment raises the important point that the typical workflows of NanoJ-Fluidics were not clear in the original manuscript. Indeed, NanoJ-Fluidics has been designed to carry protocols following a sequential total replacement of the medium of the sample, where the liquid content is first aspirated through a peristaltic pump, then changed to a new medium delivered by the syringe pumps. In this type of protocol, similar to the ones carried manually on most wet lab experiments, there is no cumulative error and only the error associated with the volume infused comes into play. This point has now been clarified in the revised manuscript by addition of Sup. Note 2: NanoJ-Fluidics workflows.

However, we agree that the characterisation of the system was lacking from the original manuscript. We therefore now include a detailed and quantitative characterisation of the errors of precision, accuracy and repeatability of the pumps, which is now described in Sup. Note 3 and Fig. S4.

The given accuracy measurements are not described in detail.

*- Was this a single syringe (for each volume) and repeated extrusions, or was it separate syringes?
- Was it a single pump? How does extrusion rate vary between pumps?*

- The motors used are direct drive and there is no encoder / feedback meaning that the amount of motion will vary depending on load [a stepper motor or continuous rotation servo would have been a much better choice here, and would not have added much to the overall cost]. Does extrusion volume depend on how much is left in the syringe? How about reagent viscosity / surface tension / liquid head? Is extruded volume linear in extrusion time?

I would like to see the reproducibility quantified:

- *in a single syringe over the full volume*
- *across syringes*
- *across pumps*
- *across different speeds and activation durations*

We agree that our characterization of the accuracy and precision of the pumps was not described in sufficient details. To address this, and following the reviewers' suggestions, we have replaced our original calibration measurements by an in-depth characterization of the LEGO syringe pumps across different pump units, syringes and injected volumes (Sup. Note 3 and Fig. S4):

- We determined that individual LEGO syringe pump units do vary in precision and accuracy: these values are reasonably negligible (well below 5% as described in Fig. S4b-c), especially for the type of workflows (Sup. Note 2, Fig. S4a) and experiments described in the manuscript, as now discussed in Sup. Note 2 & 3.

- We now describe the calibration protocol that helps minimizing the errors of accuracy and this is detailed in the methods section and discussed further in Sup. Note 3. We've added an option to the NanoJ-Fluidics software interface to calibrate the flow-rate for each individual LEGO syringe pump (Fig. S3), permitting all syringe pumps to be equally calibrated and effectively indistinguishable (Fig. S4d). It is important to note that in Fig. S4b-c, the data presented is from non-calibrated pumps and this therefore highlights that the errors of NanoJ-Fluidics are already considerably small even without calibration.

- We characterised the pump units across different syringe sizes (1, 2 and 5 mL) and determined that both accuracy and precision have errors below +/- 5% (Fig S4e).

- We determined that the syringe pumps are accurate and precise across a wide range of injected volumes, as can be observed by the fidelity of the measured values in relation to the nominal values using both a 1 mL and a 2 mL syringe with a calibrated LEGO syringe pump unit (Fig. S4f).

We agree that the absence of encoder and feedback is a limitation in terms of accuracy but as our new measurements show, the accuracy and precision obtained with NanoJ-Fluidics are sufficient for most applications (Sup. Note 2&3 and Fig. S4). Further, since the syringes are oriented horizontally the load will be mainly limited by the friction of the plunger along the barrel, which translates into an accurate and precise liquid injection independent of how much volume the syringe holds (Fig. S4c).

Importantly, the critical step for the success of the protocols presented here does not lie in the absolute volumes added to the sample but rather in the concentration of the solutions added, as the workflow entails a full replacement of the liquid (prepared at the correct concentration) in the sample in all steps (Sup. Note 2). The concentration of the added solution depends only on the accuracy of the equipment used to prepare it (e.g. research-grade pipettes) at the setup step.

For very valuable solutions, such as antibodies or specific drugs for example, the Overflow loading model can be used to minimize waste. With this in mind, by adapting the different loading modes to the protocols, we effectively render the accuracy and precision errors of injection negligible (Sup. Note 2 & 3 and Fig S4a). Regarding the disposal of medium with the peristaltic pumps, accuracy is not essential since these steps typically intend to remove the volume entirely and can be ensured by setting the removal of ~110% of the expected remaining volume.

This said, as the reviewer points out, the use of stepper motors may be advantageous in other situations and we are considering using them in future iterations of NanoJ-Fluidics (e.g. using LEGO Mindstorm NXT motors).

We have now emphasized and clarified all of the aforementioned aspects in the results section and Sup. Note 2&3. We have also created a dedicated new section in the NanoJ-Fluidics Wiki with calibration instructions: <https://github.com/HenriquesLab/NanoJ-Fluidics/wiki/Calibrating-pumps>

The Arduino code linked on Github is such that motor control and IO are performed in a sequential loop. This will put some constraints on timing accuracy and hence minimum extrusion volume / extrusion accuracy at small volumes. I would imagine that the accuracy at small volumes/times would be considerably worse than with a 10s duration. Whilst this is not a critical problem, the authors should comment on this and state where the practical limits lie.

The reviewer raises an extremely important point that was not properly characterised in the original manuscript. To address this, we determined the lower limits of the volume that can be dispensed with good precision and accuracy (Fig. S4f). We measured the accuracy of our system with the two syringes that would be used for fine volume injection, 1 and 2 mL. The fidelity of the measured values in relation to the nominal values suggest that the LEGO syringe pumps can be used reliably to inject volumes above ~ 20 μ L for both syringes. We have highlighted this in Supp. Note 3.

It would additionally be nice (and this is of relatively minor importance) if a full BOM with indicative costs were provided so as to give a quick indication of how the cost compares to commercial options.

We now provide a full breakdown of the materials of the approach in Supplementary Table S1 for a single syringe pump, an array of 4 pumps and one made of 8 pumps.

Reviewer #2 (Remarks to the Author):

General points:

...Together with the open source design, setup and software details, I expect this to be of interest to developers of microscope hardware and protocols, and to those who require such media exchanges in their imaging experiments. It is a design that I expect laboratories will consider testing or developing further, and will provide a clear advantage over manual methods (e.g. those similar to previous exchange-DNA-PAINT: Jungmann, Nat. Methods (2014), as cited) when automation of the fluid exchange is a requirement of an experiment...

We thank the reviewer for the positive feedback.

The automatic, event-triggered fixation demonstrated in a supplementary note seems the clearest example of where NanoJ-Fluidics would come into its own, if mitotic rounding and subsequent fixation are detected and triggered without user input at that point. Is that done without user input? If not, can that be done?

While the event detection on Fig. S8 is done with user input, it could also be done automatically using the NanoJ-Fluidics during acquisition, thanks to its integration with μ Manager, an ImageJ plugin for microscope control (<https://micro-manager.org>). In this case, the researcher carrying the experiment could use the μ Manager scripting interface to control the fluidic devices during the acquisition and trigger fluid exchange based on in-line image analysis. We have now made these points clearer in the methods section.

The potential to put a pump inside a sealed incubator (Fig S1) is another example of this requirement for usefulness of automation, or remote control. In other examples, automation is convenient, but perhaps less necessary, and manual use of the same syringes and tubes may suffice, although it would require more time spent at the instrument. There will be other potential applications where the automation or remote control is necessary. These may be a better demonstration of the usefulness of the NanoJ-Fluidics concept in the main paper, or other advantages not specifically discussed could perhaps be presented.

We agree with the reviewer that Fig. S1 and the corresponding experiment in Fig. S8 are key examples of the potential of the NanoJ-Fluidics and are a great showcase for the broad scope of NanoJ-Fluidics. We are also happy that the reviewer sees the many potentials of NanoJ-Fluidics, and as suggested, it would be unrealistic to present an exhaustive list of possible protocols. We did not add Fig. S8 to the main text as we wanted to focus on the benefits of using NanoJ-Fluidics in Super-Resolution experiments, Fig. S8 uses diffraction limited imaging.

We believe that the simplicity of the experiments showcased here cover a broad range of potential applications without overloading a potential reader. However, because we believe that the higher the number of different applications, the more users will be drawn to NanoJ-Fluidics, we further set up a forum (<https://gitter.im/NanoJ-Fluidics>) where users can discuss and showcase different experiments with the system. This forum is now directly linked to the NanoJ-Fluidics Wiki.

Specific points

1. Introduction

'Carrying elaborate protocols, interlacing imaging and manual media replacement, critically hampers their reproducibility...': This sentence is difficult to read clearly.

We have rephrased this sentence to improve its readability. It is now as follows:

"However, the adoption of such elaborate protocols are commonly hampered by low reproducibility and throughput, limiting their appeal for quantitative work."

2. Construction / Supplementary note 1

While LEGO is easily available, the instructions the reader is directed to on the internet indicate that not all parts are available from the same source, and sourcing the Lego Technic components may be less reliable and more complicated, from second-hand sellers. A statement or list of how many sources (vendors) of LEGO components were required, or would currently be required to build the system of 4 or 8 pumps would be informative to the potential builder. Alternatively or additionally, 3D printing models could be provided that would allow builders to acquire the construction parts that way.

The reviewer is right in pointing out that not all the LEGO parts can be sourced from the same supplier. To simplify the construction of a NanoJ-Fluidics LEGO syringe pump array we have created a predefined wish list that streamlines the ordering of the individual components. The second-hand shops highlighted in the Wiki track the availability of specific components and can be configured to send warnings when desired parts are available. This said, we have now built 6 NanoJ-Fluidics arrays composed of 8 LEGO syringe pump units each and have never faced component shortages. In connection with reviewer #1's comments we have populated a detailed list of parts in Table S1 with estimated prices and vendors for all parts. We hope that this will support replication of our work.

'polydimethylsiloxane' should not be capitalized.

This has been amended in the text.

3. Live-to-fixed super-resolution microscopy

The authors use SRRF processing to improve the resolution of their live-cell images, acquired with HILO illumination. However, the resolution comparison is confusing, since what is stated for SRRF appears to be the minimum resolution for the single region of the image with the smallest resolution over all regions and timepoints, whereas approximate HILO resolutions are provided at each timepoint. Later on, the dSTORM and DNA-PAINT images are characterised by 'minimum resolution', presumably as an indication of what is achievable using this protocol. Perhaps this 'minimum resolution' would be fairer terminology for the live-cell images, and usable for both SRRF and HILO, as well as the dSTORM image of Fig. 2. In this case, there is no need to give these figures as an approximation. If an indication of the average image resolution is desired, that would need defining differently. If the comparison between unprocessed and SRRF images needs to be made here, perhaps similar FRC maps for the unprocessed HILO image sequences could be provided as well.

We thank the reviewer for pointing out this discrepancy. We now show the FRC maps for the HILO dataset (Fig. S5) and we mention the minimum in-cell resolution across the manuscript. We also clarified the how the resolutions were estimated (in-cell, mean and min from FRC maps) in Sup. Note 4.

4. Mutliplexed STORM and DNA-PAINT

The definition of 'resolution' is clearer here, but can still be tightened by removing the approximation symbols and stating '66–68 nm minimum resolution, expect for actin... 97 nm...', seeing as this seems to be well-defined. Please confirm that these regions of minimum resolution include parts of the cell – especially for the actin image. If not, what is the minimum resolution within the cell image?

All FRC resolution values presented here have been checked to be in-cell.

Proof of the ability to multiplex super-resolution imaging is given here, but discussion of the resolutions achieved would be beneficial somewhere, since higher resolutions should be possible using NanoJ-Fluidics with alternative protocols. Probably, practical considerations will have led to the choice of ATTO488 for the actin image, which may have resulted in worse performance than the dSTORM image of actin in Fig. 2, for example. It would be good to explain this difference, and the reasons for this choice. In other images, the use of both primary and secondary antibodies will have decreased resolution compared to what is possible, for example with DNA-labelled primary antibodies (e.g. Jungmann, Nat. Methods (2014), as cited). There may be other known factors at work which would be helpful to understand. Such a discussion may then allow readers to see greater potential in the system.

The reviewer is right to highlight that the dSTORM fluorophore used here is not the classical AF647 and that the PAINT strands could have been conjugated to primary antibodies.

Regarding the choice of ATTO488 for dSTORM, it is a robustly validated choice by work published by us (Leterrier *et al.*, Cell Reports, 2015), and has the advantage to abrogate any possible ghosting when imaging with near-IR emitting DNA-PAINT imager strands.

The choice to conjugate secondary antibodies rather than primary antibodies is related with practical considerations as conjugation of primary antibodies may decrease their affinity and/or avidity. Given that the focus of the paper was not to showcase the highest possible resolution DNA-PAINT experiment but that NanoJ-Fluidics enables the automation of DNA-PAINT experiments that are easy to carry by most labs, we chose to conjugate secondary antibodies.

For reproducibility, can the authors provide more specific information about the filtering and rendering procedure applied in ThunderSTORM?

We have now added the details of the reconstruction procedure in the methods section.

5. Fig S6 / Supplementary note 3

I did not find it clear what the last sentence is about (loss of intensity in the top left of Fig S6). Is there a particular part of Fig S6 that this refers to?

NanoJ-SQUIRREL is sensitive to a wide variety of artefacts in images, and this includes differences in relative intensity across the image (certain parts of the image being brighter than others in a different way between the reference image and the super-resolved image). These intensity differences can be the result of sample preparation problems, such as fixation and/or permeabilization-induced cell morphology alterations, but also from the non-linearity of the reconstruction algorithm. We have rephrased the last sentence to reflect this.

“There is also a loss of fluorescence intensity during fixation in the top left portion of the region shown in Fig. S7, as highlighted by the large bright patch on the left-hand side of the error map.”

6. Event-driven fixation imaging (supplementary note 4)

This application is potentially a good illustration of the usefulness of the automated fluidics, and would add significance to the main text of the paper, particularly if the mitotic rounding was detected automatically and triggered fixation without user input at that point (see general points). The boxes referred to as insets look like they are simply boxes to label cells; there do not appear to be inset images.

We appreciate the reviewer comments, as mentioned above the fixation was triggered based on user input. The only box that refers to the inset is the dashed box, we have made that clear in the figure legend and changed the colour of this box.

7. Discussion

NanoJ-SQUIRREL is given twice as a reference for how NanoJ-Fluidics allows easier optimisation of experimental protocols. I think the cross-reference to Sup. Note 3, where NanoJ-SQUIRREL is used is enough, and about right, to point out the potential connection for this purpose.

Unfortunately, we could not identify the double reference the reviewer highlights. This confusion may have been due to the fact that we originally used two separate bibliography for the main manuscript and the methods. In any case, we now have a single bibliography and we checked that no double references were present.

Similar to my general point, it is stated here that NanoJ-Fluidics provides significant advantages for such combinations as dSTORM and exchange-DNA-PAINT; but these media exchanges are already ‘allowed’ by a selection of syringes without automation (e.g. similar to Jungmann, Nat. Methods (2014), as cited). It would be good to hear what particular advantages over manual use of the syringes are in view here.

We agree with the reviewer that it is possible to achieve buffer exchange manually. However, we believe that the advantage is the automation and repeatability of the complete protocol using the pumps. Our approach does not only help with the transition between dSTORM and DNA-PAINT specifically, but for all sequential labelling steps. We have clarified this point in the Discussion section as follows:

“NanoJ-Fluidics provides significant advantages in this context, by allowing, for instance, the easy automation of STORM (14) with sequential DNA-PAINT (3) protocols (Fig. 3).”

With regards to the comparison with microfluidics, concerning reagent volumes, it would be beneficial to hear how much dSTORM buffer is required to pass through the tubing in the implementation presented here.

With an inner tubing of 0.2 mm and a tubing length of 1m, the volume necessary to pass through the tube is only < 150 μ L. However, due to the modular nature of the syringe pumps, and as showcased in Figure S1, one could place a pump on the microscope stage and reduce this volume to below 15 μ L, if 10 cm long tubing was used. We added a comment in Supp. Note 2 to explicitly describe this aspect of low dead volume:

“Additionally, the modular nature and design pliability of the LEGO® syringe pumps allow to place a syringe pump on the microscope stage and easily reduce the tubing length and the corresponding dead volumes to as low as 15 μ L (estimated from a 0.2 mm inner diameter tubing and a 10 cm long tubing).

I recommend that 'robustness of the NanoJ-Fluidics' should read 'robustness of NanoJ-Fluidics', for consistency with its treatment elsewhere.

Amended.

'For example, optimal SMLM images requires fine-tuning the fixation...' needs rephrasing.

This sentence was reworded as follows:

"For example, to reach optimal SMLM images it is often necessary to fine-tune fixation, permeabilisation, blocking, antibodies concentration and imaging buffer composition (17)."

8. Methods

Live-to-fixed imaging:

For reproducibility, what power 405 nm pumping was used, when it was necessary?

The methods were amended as follows:

"(...) with 405 nm pumping when required (0.5-1% of maximum output when the blinking density was below 1 particle/ μm^2)"

Multiplexed super-resolution:

Is the 'PEM buffer' the same as the 'cytoskeleton-preserving buffer' of the previous section? If so, it would be more clearly named again as such, or named PEM buffer the first time.

The methods were amended as follows:

"(...)Cells were fixed at 37°C for 15 minutes with 4% paraformaldehyde in the cytoskeleton-preserving buffer "PIPES-EGTA-Magnesium" (PEM: 80 mM PIPES pH 6.8, 5 mM EGTA, 2 mM MgCl₂) (25)."

For reproducibility, what is the 'alternating way' in which frames are acquired for two-colour DNA-PAINT (for both I1/I2 and I3/I4)? For clarity, what do the 60% and 30% refer to? Is it laser output power, or something else? Can the authors be more precise about the concentrations for the various imager strands?

Alternating way refers to an image of the 647 nm channel alternating with one image of the 561 nm channel resulting in 30.000 frames of each channel. We have clarified that section as follows:

"After injection of the I1-ATTO655 (0.25 nM) and I2-CY3B (2 nM) imagers in buffer C, 60,000 frames were acquired in an alternating way (60% power of the 647 nm laser and 30% power of the 561 nm laser, 30,000 frames of each channel) to image TOM20 and vimentin, respectively. After three rinses with buffer C, I3-Cy3B (1 nM) and I4-ATTO655 (0.5 nM) were injected in buffer C, and 60,000 frames were acquired in an alternating way (30% power of the 561 nm laser and 60% power of the 647 nm laser, 30,000 frames of each channel) to image clathrin and microtubules, respectively".

'I3-Cy3B and I4-ATTO655 were in buffer C were injected' should read 'I3-Cy3B and I4-ATTO655 were injected in buffer C'.

Amended.

Event detection and live-to-fixed imaging: Here or in the supplementary note, more detail would be good about the recognition of the appropriate timepoint for fixation: how was the 'minimal area' (suppl. note 4) recognised as such?

We appreciate the reviewer's comment; indeed "the minimal area" is not a clear metric. Fixation was triggered when enough cells rounded, which was identified by visual inspection. We have clarified that point in Sup. Note 6, as follows:

"When enough cells were observed to undergo mitotic rounding (determined by visual inspection), the fixative was injected."

'underwent' should read 'undergone'.

Amended.

Reviewers' comments:

Reviewer #2 (Remarks to the Author):

While I see it as a positive thing that the instructions for the replication of an inexpensive and useful LEGO system like NanoJ-Fluidics are made freely available, I still believe the manuscript is lacking demonstration of the full experimental potential of the system. Control with micromanager and the flexibility that can provide has been referred to in the author's response, but not obviously in the paper, unless I have missed this. What would significantly improve the manuscript is demonstration of automated fluidics where automation (and not only user-timed remote control) is a genuine requirement of the experiment, or a clearly significant improvement to the protocol, compared with the manual use of syringes. In the experiments currently given, a user could repeatedly add reagents manually at the right times. For instance, automatically triggering the next phase of an imaging experiment and adding a reagent based on image analysis during acquisition could be illustrative (e.g. as in live-to-fixed imaging, or perhaps when blink rate has reduced to a certain level in localisation microscopy).

__Construction/Supplementary note 1__

To assist the reader, the parts list in Table S1 should be more detailed than 'See wiki' for the LEGO components. At least an indication that this will require more than a visit to the LEGO store/site is needed, if not the level of detail provided in the wiki. e.g. 'LEGO components were obtained from vendors A, B and C'. Or 'LEGO components were sourced through vendors X, Y and Z'.

__Multiplexed STORM and DNA-PAINT__

The choice of dyes and labelling technologies makes sense, but the dyes used need stating in the main text and figure legend, as done for figure 2 and its text.

In Supplementary Note 4, the actin channel is described as a DNA-PAINT channel. This needs changing.

__Supplementary note 5, last sentence; Fig. S7__

This is still unclear. Which error map from Fig. S7 is meant? All four? STORM vs. GFP live? Why is it useful that the colour bars in Figs. S7c and S7d have the same scale? Can the colour-scale of Fig. S7d be changed to better show the error pattern and changes between live and fixed?

__Supplementary note 6__

"In order to fully exploit the fluidics automation of NanoJ-Fluidics, ..." In fact, full exploitation in this case would be using image analysis of rounding cells to trigger the injection of the fixative, which would be good to see.

__Dead volume: Supplementary Notes 2 and 3, Fig. S4__

The estimation of dead volume is helpful. The Overflow loading method is also potentially helpful. However, I could not see the results of the statistical test on the measurements testing this method beyond 'ns' in Fig S4g. What exactly were the test and its results? In what sense are the measurements 'paired'? N = 5 seems low for such an assessment of accuracy. Central tendency and variability of the difference between pipetting and Overflow loading would be useful to include, once more repeats are obtained. Fig. S4h is referenced in Note 2, but there is no Fig. S4h, and this should perhaps be S4g.

__Methods: multiplexed super-resolution__

'in an alternating way' is still confusing. Better would be '30,000 frames were acquired in each channel, first at 60% power of the 647 nm laser, then at 30% power of the 561 and laser'. If that is what is meant. And similarly for I3 and I4.

__Methods: SMLM and SRRF image reconstruction__

FRC needs spelling out. Niewenhuizen, et al (2013), Nature Methods, 10, p557, would be a helpful

reference. I assume the FRC threshold is set to $1/7$, in which case just this reference is fine in the method. Is that right?

Dear Reviewer,

We thank you for the overall positive assessment of our manuscript. Below we address the standing points raised and present a revised manuscript based on your feedback.

Reviewer #2: While I see it as a positive thing that the instructions for the replication of an inexpensive and useful LEGO system like NanoJ-Fluidics are made freely available, I still believe the manuscript is lacking demonstration of the full experimental potential of the system. Control with micromanager and the flexibility that can provide has been referred to in the author's response, but not obviously in the paper, unless I have missed this. What would significantly improve the manuscript is demonstration of automated fluidics where automation (and not only user-timed remote control) is a genuine requirement of the experiment, or a clearly significant improvement to the protocol, compared with the manual use of syringes. In the experiments currently given, a user could repeatably add reagents manually at the right times. For instance, automatically triggering the next phase of an imaging experiment and adding a reagent based on image analysis during acquisition could be illustrative (e.g. as in live-to-fixed imaging, or perhaps when blink rate has reduced to a certain level in localisation microscopy).

Authors: We agree the manuscript would benefit from demonstrating a user-independent autonomous experiment requiring fluidics. We now include new data showing this capability - Sup. Note 6, Fig. S8 and Movie S3. In these datasets we show real-time image analysis detecting cell shape changes (cell rounding prior to mitosis) which automatically triggers a protocol for fixation, permeabilization, wash and post-fixation staining (Phalloidin and DAPI). Completion of the fluidics protocol is automatically followed by a second imaging session (multi-colour z-stack acquisition). This entire process was completely unsupervised and required no user input. It provides an excellent example, where the fluidics automatically fixes cells upon a biological cue, here when they reach a specific stage of cell cycle, while preserving temporal information on the behaviour of the living cells before fixation.

Reviewer #2: __Construction/Supplementary note 1__

To assist the reader, the parts list in Table S1 should be more detailed than 'See wiki' for the LEGO components. At least an indication that this will require more than a visit to the LEGO store/site is needed, if not the level of detail provided in the wiki. e.g. 'LEGO components were obtained from vendors A, B and C'. Or 'LEGO components were sourced through vendors X, Y and Z'.

Authors: This has now been changed to "LEGO™ store and Brick Owl LEGO™ marketplace". We agree that Table S1 is not showing all the details of suppliers. However, not all parts can be ordered from the LEGO store, the LEGO marketplace in turn congregates hundreds of stores throughout the world. Each user can choose the one geographically closer to them or the one that offers the best price. Therefore we are of the opinion that pointing the readers to the marketplace is the easiest way for them to source efficiently the LEGO parts necessary for NanoJ-Fluidics.

Reviewer #2: __Multiplexed STORM and DNA-PAINT__

The choice of dyes and labelling technologies makes sense, but the dyes used need stating in the main text and figure legend, as done for figure 2 and its text. In Supplementary Note 4, the actin channel is described as a DNA-PAINT channel. This needs changing.

Authors: Details on labelling used on Figure 3 has now been added:

Line 109-112 in main manuscript: “(...) *b) Left, full view showing 5-channel merge of STORM and DNA-PAINT with actin (yellow, Phalloidin ATTO488), vimentin (blue, Cy3B DNA-PAINT imager strand), β -tubulin (green, ATTO655 DNA-PAINT imager strand), clathrin (cyan, Cy3B DNA-PAINT imager strand) and mitochondria (red, ATTO655 DNA-PAINT imager strand).*”

We also changed the text accordingly:

Line 98-102 in main manuscript: “(...) *Fixed cells were labelled using primary antibodies targeting mitochondria, vimentin, microtubules and clathrin followed by DNA-coupled secondary antibodies (Cy3B labelled imager strands for vimentin and clathrin, ATTO655 labelled imager strands for mitochondria and microtubules), as well as fluorescent phalloidin ATTO488 to label actin.*”

Additionally Supplementary Note 4 now reads:

Line 161-162 in SI: “(...) *whereas for the actin STORM channel an estimated resolution of 97 nm was obtained.*”

Reviewer #2: __Supplementary note 5, last sentence; Fig. S7__

This is still unclear. Which error map from Fig. S7 is meant? All four? STORM vs. GFP live? Why is it useful that the colour bars in Figs. S7c and S7d have the same scale? Can the colour-scale of Fig. S7d be changed to better show the error pattern and changes between live and fixed?

Authors: We agree this section required some clarifications. We improved the description as follows:

Line 168-173 in SI: “(...) *There is movement of the bright filaments within the cell body and filopodia at the cell periphery on a sub-micron scale (highlighted by arrows on Fig. S7b - Merge). This degree of movement is comparable to or smaller than the frame-to-frame movement shown in Sup. movie S1. There is also a loss of fluorescence intensity during fixation in the top left portion of the region shown on the left-hand side of the difference map (Fig. S7a), as highlighted by the large bright patch (yellow, corresponding to live-cell information).*”

We have also changed Fig. S7d according to the reviewer's suggestion.

Reviewer #2: __Supplementary note 6__

“In order to fully exploit the fluidics automation of NanoJ-Fluidics, ...” In fact, full exploitation in this case would be using image analysis of rounding cells to trigger the injection of the fixative, which would be good to see.

Authors: We have now added a complete dataset highlighting the capacity for integration and automation of NanoJ-Fluidics with micro-manager in Sup. Note 6, Fig. S8 and Movie S3.

Additionally, further details are given in the discussion and materials and methods as follows:

Line 122-129 in main manuscript: *“We also show that sample treatment can be efficiently triggered on specific biological cues such as mitotic cell rounding either manually or in a fully unsupervised manner through the integration of NanoJ-Fluidics with the image acquisition (Sup. Note 6 & 7, Fig. S8 & S9 and Movie S3 & S4). These experiments showcase the broad scope of NanoJ-Fluidics in increasingly complex experimental settings and highlights its potential in the context of recently described high-throughput, high-content approaches 16,17. This paves the way towards unsupervised, high-content, event-driven correlative live-to-fixed imaging or sample treatment compatible with highly flexible experimental workflows of imaging and fluidics exchange.”*

Line 151-154 in main manuscript: *“As highlighted by our unsupervised detection of mitotic cell rounding and sample treatment (Sup. Note 6, Fig. S8 and Movie S3), our μ Manager plugin therefore allows for NanoJ-Fluidics to be fully integrated into microscopy acquisition software, enabling a seamless combination between the imaging and fluid exchange protocol.”*

Line 348-374 in methods of main manuscript: *“Unsupervised NanoJ-Fluidics workflow and integration with μ Manager. HeLa HRTG were plated on 35 mm iBidi dishes and incubated with Cdk1 inhibitor (RO-3306, 5 nM) for 15-20 hours before imaging. Immediately prior to imaging, cells were washed with fresh medium using NanoJ-Fluidics and live cell imaging was started. For each experiment, 30-50 fields-of-view were selected and imaged every 5 min in the GFP and mCherry channels. Imaging was performed at 37 °C and 5% CO₂ on a Nikon Eclipse Ti with a CFI Apo TIRF 60X Oil (1.49 NA), LED illumination (CoolLED P-4000) and Andor Zyla 4.2 sCMOS camera (100 ms exposure time and 20% illumination for each channel). A BeanShell script (available on our GitHub page) was designed to run the acquisition and image analysis in parallel. For each field-of-view cells were segmented by Otsu thresholding followed by morphological opening. The circularity of each cell was calculated and if the average circularity of a field-of-view exceeded a user-defined circularity threshold (here set to 0.82) the field-of-view was declared as triggered. The circularity was estimated via the common shape factors measurement of ImageJ, and is estimated as follows:*

$$C = \frac{4\pi A}{P^2}$$

Where C is the circularity estimator, A is the area and P is the perimeter.

Once a user-set fraction of fields-of-view triggered (here 25%), the “cell rounding event” was activated. At that stage, NanoJ-Fluidics fixation and labelling protocol were initiated (a small delay was added to visualize the evolution of the rounded cell further, here 15 min). The NanoJ-Fluidics sequence was set as follows:

- Fixation with 4% PFA in PEM buffer for 15min at 37 °C.
- Wash with 1X PBS for 10 min, (x3)
- Permeabilisation and blocking in 1X PBS with 5% BSA and 0.05% Triton-X for 30 min.
- Staining with Phalloidin-AlexaFluor647 (4 units/mL) and DAPI (1 μ g/mL) for 1h.
- Wash with 1X PBS for 10 min, (x2)

Once the fluidics steps were executed, the post-fixation/staining imaging was then automatically performed taking a 3D z-stack of 30-40 planes (spaced by 0.5 μ m) in DAPI,

GFP, mCherry, and AlexaFluor647 channels (200 ms exposure times and 20% illumination for each channel)."

The BeanShell script is made available via our GitHub page:

<https://github.com/HenriquesLab/NanoJ-Fluidics/wiki/GUI-External-Control>

As well as a detailed description of how to use the script:

<https://github.com/HenriquesLab/NanoJ-Fluidics/tree/master/beanshellScripts>

Reviewer #2: __Dead volume: Supplementary Notes 2 and 3, Fig. S4__

The estimation of dead volume is helpful. The Overflow loading method is also potentially helpful. However, I could not see the results of the statistical test on the measurements testing this method beyond 'ns' in Fig S4g. What exactly were the test and its results? In what sense are the measurements 'paired'? N = 5 seems low for such an assessment of accuracy. Central tendency and variability of the difference between pipetting and Overflow loading would be useful to include, once more repeats are obtained. Fig. S4h is referenced in Note 2, but there is no Fig. S4h, and this should perhaps be S4g.

Authors: We thank the reviewer for the positive comments. Indeed the dead volume is an important factor to have in mind, but will depend on the length and thickness of the tubing used, and it is especially important if the reagents used are valuable. To overcome this issue we suggest in the text using the overflow method. This approach entails inserting the precise volume of reagents in the tubing, then the reagent can be injected to the sample followed by air injection to guarantee that any remaining reagent in the tubing is delivered.

The method used (as well as the average and standard deviation of the measurements) is now described in Supplementary Note 3.

Line 146-153 in SI: *"Finally, to test the reliability of the overflow method, we first collected 100 μ L with a Gilson P100 pipette and weighed it to accurately define the picked volume and take into account the imprecision due to the pipetting itself. We then injected that volume into the tubing, followed by a small volume of air ($< 100 \mu$ L). The reagent was then pushed out of the tubing and onto the weighing scale using the syringe pumps, by infusing the equivalent volume to a 500 μ L injection, using a 2mL syringe. This paired weight measurement (pipetted weight vs. infused weight) was performed 10 times. A paired t-test gave a p value of 0.04 that the Overflow loading mode allows the injection of volumes ($98.96 \pm 1.3 \mu$ L) with the same accuracy as when using research-grade pipettes ($99.73 \pm 0.9 \mu$ L) (Fig. S4g)."*

Additionally:

Line 262-269 in methods of main manuscript: *"(...) For the Overflow loading mode, a volume of 100 μ L was measured with a Gilson P100 micropipette (by weighing it in an analytical scale, Sartorius Cubis®, 0.1 mg \pm 0.0003 mg). The weighed volume was inserted into the tubing and then dispensed using the pump by infusing a 500 μ L volume. By flushing out a larger volume than injected, this method ensures that the entire volume is dispensed. The dispensed reagent volume was weighed and a paired t-test was used (GraphPad Prism version 7.00 for Mac, GraphPad Software, La Jolla California USA) to estimate the statistical difference between the volume collected by the pipette and that dispensed by NanoJ-Fluidics."*

We thank the reviewer for highlighting this mistake. Indeed, the reference should have pointed towards Figure S4g and we have amended this.

Reviewer #2: __Methods: multiplexed super-resolution__

'in an alternating way' is still confusing. Better would be '30,000 frames were acquired in each channel, first at 60% power of the 647 nm laser, then at 30% power of the 561 and laser'. If that is what is meant. And similarly for I3 and I4.

Authors: We would like to clarify the confusion about how the acquisition was performed. Here we used a common multi-color acquisition mode called frame interleaving where in sequence we acquire a frame from channel 1, then channel 2, repeating this cycle 30,000 times. The total acquisition therefore contains 60,000 frames (30,000 frames corresponding to channel 1 and 30,000 corresponding to channel 2). This approach has a number of advantages for multi-color SMLM imaging, notably in terms of image registration.

We have now amended the methods to describe this.

Line 310-319 in main manuscript: "(...) First, a STORM image of phalloidin-ATTO488 was acquired in buffer C (PBS 0.1M pH7.2, 500 mM NaCl) by taking 30,000 frames at 30 ms/frame at 50% power of the 488 nm laser. After injection of the I1-ATTO655 (0.25 nM) and I2-CY3B (2 nM) imagers in buffer C, 60,000 frames were acquired by sequentially interleaving the two channels (60% power of the 647 nm laser and 30% power of the 561 nm laser) to image TOM20 and vimentin in a single acquisition, therefore providing 30,000 frames for each channel. After three rinses with buffer C, I3-Cy3B (1 nM) and I4-ATTO655 (0.5 nM) were injected in buffer C, and 60,000 frames were similarly acquired via frame interleaving (30% power of the 561 nm laser and 60% power of the 647 nm laser, 30,000 frames of each channel) to image clathrin and microtubules, respectively."

Reviewer #2: __Methods: SMLM and SRRF image reconstruction__

FRC needs spelling out. Niewenhuizen, et al (2013), Nature Methods, 10, p557, would be a helpful reference. I assume the FRC threshold is set to 1/7, in which case just this reference is fine in the method. Is that right?

Authors: We have now added the citation to the main manuscript in addition to the SI and clarified the FRC analysis in the method section according to the reviewer's suggestion.

Line 344-346 in main manuscript: "The image resolution was estimated by calculating the Fourier Ring Correlation (FRC)²⁸ with the typical 1/7 threshold using NanoJ-SQUIRREL plugin and reconstructing the original dataset separated into two different stacks composed of odd or even images¹³."

REVIEWERS' COMMENTS:

Reviewer #2 (Remarks to the Author):

The demonstration of the potential of this experimental system is much improved by the inclusion of the automatically triggered sample treatment upon mitotic rounding. I would question whether this (and Fig S9) should wait until the discussion to be referred to. But that is perhaps down to editorial discussions.

Revision of statistics in Suppl. Note 3 and Fig. S4:

The method behind the paired measurements (Fig. S4g) is now clear. There is now confusion here though, in that the authors found a statistical difference at $p = 0.04$, between overflow and pipette loading, but appear to have still generalised this to 'ns' in the figure (S4g). Perhaps the (small) size and variation (standard deviation) in the difference between the two measurements could simply be described instead.

This may affect description in the Methods section as well.

The definition of the confidence intervals should be understandable throughout Suppl. Note 3 (including the first paragraph), and the t-statistic should be reported (if retaining the t-test). Error bars should be defined in Fig. S4.

If wanting to display 'Paired correlation' or similar in Fig. S4g, more than just the scatter of each measurement would be better. Perhaps lines joining the paired measurements. But again, simply displaying, or perhaps even just stating - without Fig. S4g - the (estimated) distribution of the difference between the two measurements may be sufficient.

I would recommend the manuscript be published once the issues above are resolved.

Dear Reviewer,

We thank you for the positive assessment of our manuscript. Below we address the standing points raised and present a revised manuscript based on your feedback.

Reviewer #2: The demonstration of the potential of this experimental system is much improved by the inclusion of the automatically triggered sample treatment upon mitotic rounding. I would question whether this (and Fig S9) should wait until the discussion to be referred to. But that is perhaps down to editorial discussions.

Authors: We have now migrated the description and discussion of the event-driven experiments to the main manuscript, together with the two associated figures – Fig 2 & 3.

Reviewer #2: Revision of statistics in Suppl. Note 3 and Fig. S4: The method behind the paired measurements (Fig. S4g) is now clear. There is now confusion here though, in that the authors found a statistical difference at $p = 0.04$, between overflow and pipette loading, but appear to have still generalised this to 'ns' in the figure (S4g). Perhaps the (small) size and variation (standard deviation) in the difference between the two measurements could simply be described instead. This may affect description in the Methods section as well.

If wanting to display 'Paired correlation' or similar in Fig. S4g, more than just the scatter of each measurement would be better. Perhaps lines joining the paired measurements. But again, simply displaying, or perhaps even just stating - without Fig. S4g - the (estimated) distribution of the difference between the two measurements may be sufficient.

Authors: We thank the reviewer for pointing out that inconsistency. We have followed the reviewers' suggestion and replaced Suppl. Fig. 4g by the distribution of the difference between the two measurements.

We have amended the methods accordingly, and supplementary note 3 to state:

“Comparing the overflow method ($98.96 \pm 1.3 \mu\text{L}$, where error is calculated as the standard deviation across $N=10$ repeats) and the use of research-grade pipettes ($99.73 \pm 0.9 \mu\text{L}$, where error is calculated as the standard deviation across $N=10$ repeats) resulted in a mean of difference between sample means ($\mu\text{M1-M2}$) of $0.77 \mu\text{L}$ with a standard error (SE) of $0.48 \mu\text{L}$, demonstrating the accuracy equivalence of the methodologies.”

Reviewer #2: The definition of the confidence intervals should be understandable throughout Suppl. Note 3 (including the first paragraph), and the t-statistic should be reported (if retaining the t-test). Error bars should be defined in Fig. S4.

Authors: We thank the reviewer for highlighting this.

We have amended Suppl. Note 3 to state:

“To compare the dispensed volume and the determined weight ordinary one-way ANOVA was used (using a confidence interval of 99%).”

We have amended the legend of Suppl. Fig. 4 to state:

“(…) b) Accuracy and precision measurement across (non-calibrated) pumps. The left panel shows all data from 5 reloads on 3 independent pumps ($N=10$ repeats per re-load). The right panel shows a collated data from all reloads ($N=50$ repeats per pump, $p<0.001$ with 99% confidence interval). (...) All the graphs display as lines the mean plus or minus the standard error of the mean (SEM) except in f) where the lines correspond to the mean plus or minus the standard deviation (SD).”

We have amended the methods to state:

“The dispensed volume was weighed and an ordinary one-way ANOVA was used (GraphPad Prism version 7.00 for Mac, GraphPad Software, La Jolla California USA) to estimate the statistical difference between the different conditions used in figure S4 (using a confidence interval of 99%).”

Reviewer #2: I would recommend the manuscript be published once the issues above are resolved.

Authors: We again thank you for the positive assessment of our manuscript.